



**HONO Measurement by Differential Photolysis**
**C. Reed[1], C. A. Brumby[3], L. R. Crilley[4], L. J. Kramer[4], W. J. Bloss[4], P. W. Seakins[2,3], J. D.**
**Lee[1,2], L. J. Carpenter[1]**
[1] Wolfson Atmospheric Chemistry Laboratories, Department of Chemistry, University of
York, Heslington, York, YO10 5DD, United Kingdom
[2] NCAS, School of Earth and Environment, University of Leeds, Leeds, LS2 9JT, United
Kingdom
[3] School of Chemistry, University of Leeds, Leeds, LS2 9JT, United Kingdom
[4] School of Geography, Earth and Environmental Sciences, University of Birmingham, B15
2TT, United Kingdom.
Correspondence to: James.Lee@york.ac.uk
**Abstract**
Nitrous acid (HONO) has been quantitatively measured *in-situ* by differential photolysis at 385
and 395 nm and subsequent detection as nitric oxide (NO) by the chemiluminescence reaction
with ozone ($O_3$). The technique has been evaluated by FT-IR to provide a direct HONO
measurement in a simulation chamber, and compared side-by-side with a LOng Absorption Path
Optical Photometer (LOPAP) in the field. The $NO/O_3$ chemiluminescence technique is robust,
well characterized and capable of sampling at low pressure whilst solid-state converter
technology allows for unattended *in-situ* HONO measurements in combination with fast time
resolution and response.
**1  Introduction**
Nitrous acid (HONO) is a major source of hydroxyl (OH)  radicals in the boundary layer
(Elshorbany et al., 2008; Kim et al., 2014; Levy II, 1973). HONO can be formed homogeneously
through reaction of nitric oxide (NO) with OH, heterogeneously through several pathways, or
emitted directly (Kleffmann, 2007; Lammel and Cape, 1996; Spataro and Ianniello, 2014; Su et
al., 2011). HONO is formed heterogeneously on surfaces through the reaction of $NO_2$ with $H_2O$
(Bröske et al., 2003). This heterogeneous formation of HONO is a net source of OH radicals in



the troposphere and is an important mediator of air quality, particularly in polluted environments
(Finlayson-Pitts et al., 2003; Gutzwiller et al., 2002; Lee et al., 2015). Direct emission of HONO
through vehicle exhaust is also thought to be a source (Kirchstetter et al., 1996; Kurtenbach et
al., 2001). Emission from snowpack has also been documented (Beine et al., 2008; Zhou et al.,
2001) and more recently biogenic sources of HONO have been identified from nitrite producing
bacteria (Oswald et al., 2013; Su et al., 2011), and soil crusts (Weber et al., 2015).
In urban areas HONO can be the major net source of OH (discounting radical cycling driven by
e.g. NO), contributing up to 80% of daytime OH production in winter and 50% in summer
(Elshorbany et al., 2008; Kleffmann, 2007; Villena et al., 2011b). However, the sources of
HONO and the many processes by which it forms are not well understood (Kleffmann et al.,
2006; Sörgel et al., 2011; Spataro and Ianniello, 2014; Villena et al., 2011a). There is a clear
need for *in-situ* measurement of HONO in order to better understand its chemistry and
emissions.
Currently, methods of detecting HONO are either remotely through DOAS (Febo et al., 1996;
Hendrick et al., 2014; Stutz et al., 2010), or by filter/denuder sampling (Acker et al., 2005, 2006;
Febo et al., 1993, 1996; Ianniello et al., 2007). A variety of *in-situ* techniques exist, namely:
Quantum Cascade-Tuneable Infrared Laser Differential Absorption Spectrometer (QC-TILDAS)
(Lee et al., 2011); Ion Drift Chemical Ionization Mass Spectrometer (ID-CIMS) (Levy et al.,
2014); Ambient Ion Monitor - Ion Chromatography (AIM-IC) (Markovic et al., 2012;
Vandenboer et al., 2014); Stripping-Coil Visible Absorption Photometry (SC-AP) (Ren et al.,
2011); Negative-Ion Proton-Transfer Chemical Ionization Mass Spectrometry (NI-PT-CIMS)
(Roberts et al., 2010); Incoherent Broadband Cavity Enhanced Absorption Spectroscopy
(IBBCEAS) (Pusede et al., 2014); dedicated commercial on-line, *in-situ* measurements include
Dual Laser – Quantum Cascade Laser (Aerodyne Research) and, as used in this study, Long Path
Absorption Photometer (LOPAP) (Heland et al., 2001). LOPAP has been characterized quite
extensively by other authors e.g. (Clemitshaw, 2004; Kleffmann and Wiesen, 2008; Kleffmann et
al., 2006, 2013; Ródenas et al., 2013).



Here, we demonstrate the exploitation of a known HONO interference for photolytic $NO_2$ conversion systems (Pollack et al., 2011; Ryerson et al., 2000; Sadanaga et al., 2010, 2014; Villena et al., 2012), to provide a simple photolytic technique for quantitative analysis of HONO.

## 2    Experimental

The differential photolytic HONO technique, henceforth referred to as pHONO, was developed from an existing fast $NO_x$ analyser described in section 2.1. The photolytic converter is described specifically in section 2.2.  Calibration is described in 2.3.

### 2.1    Differential Photolysis instrument

Measurement were performed using a dual channel Air Quality Design Inc. (Golden, Colorado, USA) instrument equipped with a UV-LED based photolytic $NO_2$ converter – commonly referred to as a Blue Light Converter (BLC) as described in Reed et al. (2015).

Briefly, two NO chemiluminescence analysers operate in parallel with duplicated independent equipment. The analysers share a common inlet allowing for parallel calibration of each channel. One channel is equipped with a photolytic $NO_2$ converter so that $NO_x$ can be determined with that channel whilst also measuring NO concurrently. This allows for fast (1 Hz or greater) determination of NO and $NO_2$.

In order to be able to also measure HONO, the $NO_x$ channel was redesigned so that the photolytic converter (section 2.2) operates in a switching mode. That is, the two lamps of different wavelengths operate alternately on a 50% duty cycle. Practically, the lamps switch every 30 seconds allowing for ca. 1 minute time resolution data.

### 2.2    $NO_2$/HONO photolytic converter

Photolytic converters were based on those supplied by Air Quality Design and manufactured according to their proprietary standards (Buhr, 2004, 2007) and are described in Reed et al., (2015). Practically, two UV-LED arrays are positioned at opposing ends of a cavity which is highly reflective to UV. Sample gas is introduced at one end of the illuminated cavity, exiting at the other. NO in the sample exiting the converter is enhanced over the original by photolysis of $NO_2$ or HONO, thus by calibration of the conversion efficiency these can be quantified.





Modifications were made to the control of the UV-LED elements to allow independent switching
of the lamps. The wavelength of one lamp was changed from standard (395 nm) to 385 nm in
order to overlap better with the HONO absorption spectrum, while the actual UV-LEDs (3 watt,
LED Engin, Inc.) are more efficient and higher powered than those used in previous work (Reed
et al., 2015).
The volume of the illuminated sample chamber is 16 mL which, with a standard flow rate of 1
standard L per min$^{-1}$ gives a sample residence time of 0.96 seconds at standard atmospheric
temperature and pressure (SATP). The $NO_2 \rightarrow NO$ conversion efficiency of the standard BLC
with the sample flow of 1 standard L per min$^{-1}$ was ~89 % with both lamps illuminated.
Individual lamp conversion efficiencies were 72.9 and 81.3 % ±0.1 for the 385 and 395 nm
lamps respectively. Determination of the conversion efficiency is detailed in section 2.4.
**2.3    Characterisation**
Spectral radiograms of the UV-LEDs output were obtained using the same procedure and
equipment described in Reed et al., (2015) using an Ocean Optics QE65000 spectral radiometer
coupled to a $2\pi$ quartz collector within a light sealed chamber.
Figure 1 shows the measured spectral emission of two UV-LED units of two different
wavelengths; 385 and 395 nm. Also shown is the absorption cross-section of HONO, $BrONO_2$,
and the $NO_2$ quantum yield (Sander et al., 2006). It is clear that there is greater overlap,
calculated to be 30%, of the HONO absorption features with the 385 nm LED than at 395 nm. In
R2 we see that NO is produced stoichiometrically through the photolysis of HONO. In this way,
illuminating an air sample at either wavelength yields a signal, we shall denote as $NO_2^{\dagger}$; which
represents the sum of contributions from $NO_2$ and HONO (R1 + R2) in differing proportions
depending upon wavelength.
$NO_2 + h\nu(<410nm) \rightarrow NO + O(^3P)$                                     (R1)
$HONO + h\nu(<390 \text{ nm}) \rightarrow NO + OH$                                 (R2)
The difference in $NO_2^{\dagger}$ signal measured at 385 and 395 nm corresponds to the difference in
conversion efficiency of HONO and $NO_2$ between the two wavelengths. Differences in $NO_2$



conversion efficiency of each lamp may be readily calibrated for and so taken into account (see
section 2.4). The difference in $NO_2^{\dagger}$ signal measured at 385 and 395 nm can therefore be used to
calculate the HONO present in the sample Eq. (1);

$$\frac{NO_2^{\dagger}{}_{385} - NO_2^{\dagger}{}_{395}}{HONO\ CE_{385} - HONO\ CE_{395}} = [HONO] \qquad\qquad (1)$$

Apparent HONO conversion efficiency (CE), HONO $CE_{385}$ – HONO $CE_{395}$, is determined
experimentally as described in section 2.4.
It is noted that at both 385 nm and 395 nm there is potential interference from $BrONO_2$ (or in fact
any other compounds which photolyse to give NO at either wavelength), with similar spectral
overlap (Figure 1). Assuming a quantum yield of 1 integrated over all wavelengths for $BrONO_2$,
21.5 ppt of $BrONO_2$ at 385 nm and 18.1 ppt at 395 nm would be required to produce a 1 ppt
error in the $NO_2$/HONO signal.  Due to the low abundance (< 10 pptV) of $BrONO_2$ in the lower
atmosphere (Yang et al., 2005), interference is therefore likely to be minimal (Pollack et al.,
2011). The difference in conversion for the different lamps equates to a maximum error in
HONO determination of 3.4 % [$BrONO_2$]; typically much less than 1 ppt.
The NO + OH back reaction after an air sample has exited the photolytic converter, but before
entering the high vacuum of the analyser, causing a decrease in signal from HONO is discussed
in Sec. 2.4.
**2.4    HONO and $NO_2$ Conversion Efficiencies**
The $NO_2$ – HONO converter system was calibrated for both $NO_2$ and HONO conversion
efficiency. $NO_2$ conversion efficiencies were determined following the procedure outlined by
Lee et al., (2009). The sensitivity of a detector in counts per second per part per trillion (cps/ppt)
is determined by adding a 7.5 mL min$^{-1}$ mass flow controlled flow (MFC) of NO calibration gas
(4.78 ppm NO in $N_2$, BOC) to the inlet of the analyser whilst sampling an overflow of zero air
free from $NO_x$, VOC and ozone. This equates to a calibration concentration of 12.5 ppbV NO per
channel. Zero air was generated by scrubbing dried (-40 $T_d$) compressed air using Sofnofil
(Molecular Products) and activated charcoal (Sigma Aldrich) traps. As described by Reed et al.,
(2015) this combination results in the lowest $NO_2$ signal. The sensitivity was found to be ~ 6.8



and ~ 6.4 (±5%) cps/ppt for the NO and NO$_x$ channels, respectively. In order to determine the
NO$_2$ converter efficiency a portion of the NO added to the inlet is first titrated to NO$_2$ by reaction
with ozone, typically generating 10.0 ppbV NO$_2$. Ozone is generated by illuminated a small flow
(~10 mL min$^{-1}$) of O$_2$ with a broad output low pressure mercury UV lamp (BHK Inc.) The
analyser signal (photomultiplier counts in Hz) is then recorded with neither UV-LED
illuminated, and then with each illuminated in turn to determine the increase in signal arising for
each lamp. The conversion efficiency (CE) is then determined as in Eq. (2).
$$CE = 1 - \frac{Signal_{Untitrated} - Signal_{Illuminated}}{Signal_{Untitrated} - Signal_{Titrated}} \qquad (2)$$
The NO$_2$ conversion efficiency was determined to be 72.9 ($j$ = 1.3 s$^{-1}$) and 81.2 % ($j$ = 1.7 s$^{-1}$)
±0.1 for the 385 and 395 nm lamps, respectively.
Calibration for HONO was achieved by sampling a permeation source over a range of dilutions
using methods modified from Taira and Kanda, (1990) and Febo et al., (1995). Nitrous acid was
generated by the reaction of hydrochloric acid with sodium nitrite salt as described by Febo et
al., (1995) shown in reaction 3.
HCl + NaNO$_2$ → HONO + NaCl                                       (R3)
In order to achieve a continuous source of HONO, a permeation tube (Kin-Tek, HRT-010.00-
BLANK/U) was filled with HCl (37%, Fluka, AR grade) and placed in a thermostated (30 to 55
$^o$C) permeation oven (Kin-Tek, 585) with NaNO$_2$ salt (Fluka, AR grade). The permeation oven
was flushed with 1.5 standard L min$^{-1}$ zero air. The reaction is limited by HCl which permeates at
a low rate thus allowing low concentrations (<50 ppb) of HONO to be generated continuously.
As side products of reaction 3 can also be produced, the output of the permeation source was
continuously analysed for impurities. In reaction 4 NO and NO$_2$ can be formed by the gas phase
self-reaction of HONO. In reaction 5, HNO$_3$ can be formed by reaction between adsorbed and
gas phase HONO.
2HONO$_{(g)}$ → NO + NO$_2$ + H$_2$O                               (R4)
HONO$_{(ads)}$ + NO$_2$ → HNO$_3$ + NO                            (R5)


To quantify HONO without any direct measurement and close the nitrogen balance, NO, $NO_2$,
and total $NO_y$ (NO + $NO_2$ + other reactive oxidised nitrogen species such as $HNO_3$, HONO,
PAN) were measured continuously. The differential photolysis instrument itself was used to
quantify the NO. $NO_2$ was measured directly by Cavity Attenuated Phase Shift (CAPS)
spectroscopy (Kebabian et al., 2005, 2008) using an EPA certified Teledyne API T500U, to
avoid any HONO interference (which would have been present in a photolytic measurement).
Total $NO_y$ was quantified using a Thermo Environmental 42c TL $NO_x$ analyser equipped with a
molybdenum catalytic converter which has been shown to quantify $NO_y$ species such as HONO
and $HNO_3$ (Clemitshaw, 2004; Fehsenfeld et al., 1987; Villena et al., 2012; Williams et al.,
1998). The TEI 42c TL and Teledyne API T500U were calibrated either directly with an NO
standard or by gas phase titration of NO to $NO_2$ using a Monitor Europe S6100 Multi Gas
Calibrator. Production of $HNO_3$ (R5) would be indicated by an enhancement in NO over $NO_2$, as
NO and $NO_2$ are produced stoichiometrically through the self-reaction of HONO (R4), whereas
$HNO_3$ production consumes $NO_2$ and produces NO. Thus, $HNO_3$ can be indirectly quantified by
the NO: $NO_2$ ratio, and was found to be a minimal contribution to total $NO_y$. As such, HONO
can reasonably be presumed to be equivalent to $[NO_y] – ([NO] + [NO_2] + [HNO_3])$. Measured
quantities are shown in table 1.
The stability of the HONO permeation source was recorded over a 12 hour period using $NO_x$
measured by the differential photolysis analyser (the most sensitive measurement available) as a
proxy for NO, $NO_2$, and HONO. The stability was found to be ±0.01 ppb h$^{-1}$, with a standard
deviation of 0.4 ppb. The uncertainty in the HONO source is determined by a combination of the
accuracy of the NO, $NO_2$, and $NO_y$ measurements and their respective calibrations. The NO
calibration uncertainty, due to MFC flows and standard gas accuracy is 5%, similarly for the
CAPS $NO_2$ and Thermo 42i TL $NO_y$. This results in an overall uncertainty in [HONO] of 8.7%.
In Fig. 2 the observed conversion of HONO, that is the difference between HONO conversion by
the 385 and 395 nm lamps, is shown. As can be seen HONO conversion is consistently 6.54 ±
0.21 % more at 385nm than 395 nm. The fact that the 'apparent HONO conversion' (HONO
$CE_{385}$ – HONO $CE_{395}$ in Eq. 1) is constant as a function of HONO means that the determination of





[HONO] should be a linear function of the difference in $NO_2^{\dagger}$ signal at 385 and 395 nm. This
apparent HONO conversion determines the limit of detection, which is the ability of the analyser
to discriminate the difference in signal arising from photolysis at the two different wavelengths
from photon counting noise. With an apparent conversion of 6.54 ± 0.21 % the LOD with a
sensitivity of 6.4 cps/ppt is 40 ppt min$^{-1}$. The uncertainty in the apparent conversion is a
combination of the uncertainty in the HONO source, and in the $NO_2$ conversion efficiencies of
the two lamps. This results in an overall uncertainty of 12.2%.
The effect of the back reaction of OH + NO, reforming HONO, before detection of NO, thus
reducing the NO signal in the $NO_x$/HONO measurement in the presence of HONO was
calculated using a box model in FACSMILE kinetic modelling software (MCPA Software Ltd.).
Kinetic data for $O_x$, $HO_x$, and $NO_x$ reactions taken from IUPAC Evaluated Kinetic Data
(Atkinson et al., 2004). The residence time between an air sample exiting the photolysis cell and
entering the high vacuum of the NO analyser through the ~ 25 cm of ¼ inch PFA tubing is 0.11
s. The air sample is a mixture of mostly NO, $O_3$, OH, and unconverted $NO_2$. The absence of UV
irradiation results in chemistry analogous to night-time $NO_x$ chemistry with the addition of a
significant OH source. The box model was initiated with NO, $NO_3$, $O_3$, and OH concentrations
calculated to be at the outlet of the photolysis cell at each of the eight calibration points shown
previously.  The interference from the OH + NO reaction was determined as the decrease in
[NO] during the 0.11 s residence time as a percentage of measured [HONO]. The discrepancy
was calculated to vary linearly with [HONO] from -0.97 to -2.10 %, with differences between
lamps well within the accuracy of the calibration. The degree of interference from OH in $NO_2$
and HONO determination was found to be a function of $k([OH]+[NO])$ on the timescale here
(0.11 s). Reducing the residence time after the photolysis cell would reduce the error in HONO
and $NO_2$ (in the presence of HONO). Conversely, a system with a suitably long residence time
between the photolysis cell and detector may experience little-to-no HONO interference as the
OH + NO back reaction begins to dominate. There is of course a trade off in that the data must
be corrected for ambient ozone affecting the NO:$NO_2$ ratio. It is important to note that there can
never be any negative interference in $NO_2$ caused by the presence of HONO, only positive or
none.





Outside of calibration the effect of the OH back reaction with NO is likely to be less significant
due to the presence of volatile organic compounds (VOCs) which also react with OH with
comparable rates to NO. It is therefore difficult to know the absolute HONO conversion of each
UV-LED without very accurate OH reactivity/VOC concentration measurements. Due to these
unknowns, it would not be possible to correct the $NO_2$ signal for HONO interference as might be
hoped.

## 3    Results and discussion

The pHONO instrument was evaluated in an atmospheric simulation chamber (section 3.1) and
compared in the field side-by-side with LOPAP (section 3.2).

### 3.1    Chamber measurements

The Highly Instrumented Reactor for Atmospheric Chemistry (HIRAC) is a simulation chamber
facility based at the School of Chemistry, University of Leeds (Glowacki et al., 2007a). HIRAC
is a cylindrical stainless steel chamber with a total volume of ~2.25 m$^3$, containing four fans for
mixing throughout the chamber, and with a total mixing time of ~60 s. The stainless steel
structure of HIRAC allows for pressure dependent experiments to be carried out, over the range
of ~10 – 1000 mbar. Numerous sample ports are located around the chamber allowing the
attaching of instruments or introduction of gas. A multi pass Fourier Transform - Infrared (FT-
IR) instrument (Bruker IFS/66, 128.52 m path length) is present to allow spectra of the gas
within the chamber to be taken (Glowacki et al., 2007b). HIRAC is also capable of operating
over a range of temperatures (-40 to 70°C).
Experiments were carried out at ambient temperature (20 $^o$C) and pressure (1000 mbar), whilst
the chamber was kept dark. HIRAC was filled with 80 % $N_2$ (BOC, UHP, 99.998 %) and 20 %
$O_2$ (BOC) before HONO was synthesised external to the chamber following a modified
procedure described previously by (Taira and Kanda, 1990). A 1 % aqueous sodium nitrite
solution was added dropwise to a 30 % aqueous solution of sulfuric acid. The resulting reaction
(R6) produces HONO, which was added directly to the chamber via a continuous flow of $N_2$
over the reaction mixture. This is analogous to the permeation source however, side products
need not be considered due to the direct HONO measurement afforded by FT-IR.




$2NaNO_2 + H_2SO_4 \rightarrow 2HONO + Na_2HSO_4$                                                    (R6)
FT-IR spectra were taken at 60 second intervals with a spectral resolution of 1 cm$^{-1}$, whilst the
differential photolysis analyser sampled from the chamber. Dilution of the HONO, NO, NO$_2$
mixture was achieved by partial evacuation of the chamber and subsequent refilling with
synthetic air (N$_2$/O$_2$). The average HONO concentration determined from the average of two
distinct absorbance lines at 1264 cm$^{-1}$ (*trans*-HONO, Q-branch) and 853 cm$^{-1}$ (*cis*-HONO, Q-
Branch) in the FT-IR using absorptivity data taken from University of Wuppertal internal FT-IR
cross-section database, courtesy of I. Bejan via personal communication. The absorptivity data
were $7.60 \times 10^{-4} \pm 2.90 \times 10^{-5}$ ppm$^{-1}$ m$^{-1}$ (1264 cm$^{-1}$, *tans*-HONO) and $5.48 \times 10^{-4} \pm 2.60 \times 10^{-5}$ ppm$^{-1}$
m$^{-1}$ (853 cm$^{-1}$, *cis*-HONO). Some of the spectra used in quantification are shown in Fig. 3.
Figure 4 shows the strong, positive correlation between the HONO measured by differential
photolysis and by FT-IR within the HIRAC chamber up to ~ 150 ppbV, deviating at higher
mixing ratios.
Figure 4 shows that at lower HONO mixing ratios, < 150 ppb, there is better agreement between
the pHONO and FT-IR measurements, whereas the response of the differential photolysis
technique appears to be suppressed at high [HONO]. This is a result of how a photolytic
converter operates as expressed by Eq. (3) (Ryerson et al., 2000). Here $t$ is the residence time
within the photolysis cell and $k$[Ox] is the concentration and rate constant of any oxidant that
reacts with NO. Typically this would be ozone, however, OH formed from HONO photolysis
must also be considered.
$CE = \left[\dfrac{jt}{jt+k[\text{Ox}]t}\right]\left[1 - exp^{(-jt-k[\text{Ox}]t)}\right]$                                          (3)
Having two LEDs with different HONO absorption overlap results in two values for $j$(HONO).
Using the $j$(NO$_2$) values already found (1.3 & 1.7 s$^{-1}$) as an easily determined proxy for $j$(HONO)
the change in conversion with oxidant concentration can be approximated.
Figure 5 shows how the percentage conversion of any precursor that dissociates to NO, in this
case HONO and NO$_2$, changes with increasing oxidant concentration. In the case of O$_3$ the total
conversion decreases linearly with increasing [Ox], whilst the difference between the two



remains constant (9%). Conversely, with OH, conversion decays exponentially in total, and as a
difference between two LEDs of different $j$. This effect can be seen clearly above 150 ppbV
HONO in Fig. 4. Below 150 ppbV a constant difference in conversion of 6.54% is a reasonable
approximation.
The high HONO mixing ratios within HIRAC, necessary to be detected by FT-IR (LOD ~ 40
ppb), were several orders of magnitude higher than would be expected in the atmosphere where
ppt (Beine et al., 2006; Ren et al., 2010; Zhang et al., 2009, 2012) to low ppb (Acker et al., 2006;
Febo et al., 1996; Hendrick et al., 2014; Kanaya et al., 2007; Stutz, 2004) are typical. Thus, this
non-linearity at high [HONO] is unlikely to pose a serious limitation of the differential
photolysis method, with the possible exception of areas with very high $NO_x$ backgrounds. This
could be partially mitigated by having greater photolysis power at 385 nm, in combination with
moving to shorter wavelengths with better overlap with the HONO absorption cross-section. It is
clear in Fig. 1 that the 385 nm UV-LED has significantly lower light output than at 395 nm; this
is reflected in their respective $NO_2$ conversion efficiencies (72.9 and 81.3%). Alternatively,
separate 385 and 395 nm converters can be employed working in parallel, thus doubling the
number of UV-LEDs and doubling the photolysis power at each respective wavelength. This
would also allow for fast measurement simultaneously i.e. 1 Hz or faster. Alternatively, the
lower conversion efficiency at high [HONO] could be calibrated for, though as shown in the
following section, in typical atmospheric conditions no calibration or correction was required.
**3.2    Field measurements**
The Weybourne Atmospheric Observatory (Penkett et al., 1999) is a regional GAW station
located on the North Norfolk coast, UK (52°57'01.5"N 1°07'19"E). The WAO has a long history
of atmospheric measurements stretching back to its inception in 1994. During summer 2015, the
WAO was host to the Integrated Chemistry of Ozone in the Atmosphere (ICOZA) campaign,
ostensibly measuring ozone production rates. As part of the campaign a Long Path Absorption
Photometer (LOPAP-03, QUMA GmbH) (Heland et al., 2001) was deployed in order to measure
HONO. Alongside the LOPAP, the NO, $NO_2$, HONO (Differential photolysis) instrument
described in section 2.1 measured concurrently at a 1 minute time resolution.



During the ICOZA campaign, a high variation of HONO concentrations (up to ~ 500 ppt) was
observed by the LOPAP on the 1$^{st}$ and 2$^{nd}$ of July providing an ideal opportunity for comparison
between the two methods. The pHONO was deployed with replacement UV-LEDs with greater
output.  Both the 385 and 395 nm lamps had the same photon flux, indicated by identical $NO_2$
conversion efficiencies (~ 89%), in the expectation that better HONO conversion, and therefore
sensitivity, would be achieved. The estimated increase in overlap with the HONO adsorption
spectrum of the new 385 nm LED was 45% compared to 30% calculated for the original LED.
Thus lamps were installed as-is without calibration to mitigate the fall in output over time that
affects the LEDs, particularly the 385 nm LED. The decreasing output is believed to be a result
of the power control circuitry of the LEDs which does not limit the current draw immediately
after power is supplied, only after a few seconds. This means every time the lamp is switched on
it outputs its maximum (with corresponding heat), which, when used in a 30 s$^{-1}$ switching mode
as here shortens the life considerably.
The pHONO instrument sampled from an inlet box (also housing a $NO_y$ converter) located ~ 4 m
from ground level on the sampling tower at Weybourne. The sample point was connected to the
instrument by a 12 m ¼ inch PFA line (Swagelok) which was shared by the CAPS $NO_2$
instrument, thus the flowrate was ~ 3 standard L min$^{-1}$, resulting in a residence time of ~ 3
seconds. The LOPAP instrument, which has its own inlet, sampled from the roof of a specially
converted van located 20 m away upslope. Consequently, both instruments sampled at a similar
height and there was clear, unobstructed line-of-sight between them. The pHONO inlet was only
~1 m above the Weybourne observatory roof which may have contributed to the turbulent
dynamics observed in the data. The pHONO instrument was calibrated for sensitivity in ambient
air twice nightly at 00:00 and 04:00 am; NO offset was taken between these times. $NO_2$
conversion efficiencies were determined in zero air once per week. Limits of detection were 1.5
ppt min$^{-1}$ and 1.9 ppt min$^{-1}$ for NO and $NO_2$, respectively. The LOPAP was operated and
calibrated according to the standard procedures described in Kleffmann and Wiesen, (2008), with
a detection limit of  3 pptV and time resolution of 5 minutes. Zero measurements using high
purity $N_2$ (N5 grade, BOC) were performed every 12 hours on the LOPAP.
Figure 6 shows the HONO time series from both the LOPAP and pHONO instruments during
three days of high HONO measurements.



There is reasonable agreement between the established LOPAP method of HONO measurement
and that provided by the pHONO instrument without correction or calibration (Fig. 6). During
the high ozone and high HONO events observed on the 1ˢᵗ and 2ⁿᵈ especially there is very good
agreement between the two. Gaps in the data represent times where the pHONO limit of
determination was reached; where there are too few points in the averaging window after
statistical analysis of the data to be meaningful. This is because in real atmospheric conditions
the pHONO instrument is hampered by the time resolution that data is collected i.e. if there is
strong turbulence, meaning the $NO_2$ or HONO concentration varies rapidly on a timescale
shorter than that at which data is collected, then wide scatter is observed as was the case at
Weybourne.  Strong boundary layer transport meant that $NO_2$ measurement varied up to 1.5 ppb
in a minute. This is because of the way the data must be processed by interpolating between
measurements and subtraction of the 395 nm signal form the 385 nm signal. Decreasing the time
between photolysis switching (from 30 s) would obviously decrease this effect, but ultimately,
separate 385 nm (or lower) and 395 nm analyser channels are required. Consequently the data
analysis routine for the pHONO data includes tests for the variability of the data, discarding
points which show >5% variation from the subsequent point. Data failing this test is discarded
and results in gaps; this is the effective limit of determination. The data is then treated with a
robust-LOESS (Cleveland, 1979) algorithm to remove extreme values. The gaps in the time
series of LOPAP (Fig 5) were due to the removal of zero measurements and false spikes due to
bubbles passing the detector.
Figure 7 demonstrates the level of agreement in the measured HONO concentration by the
LOPAP and pHONO methods from 1ˢᵗ and 2ⁿᵈ July. From Fig. 7, the observed correlation ($r^2$ of
0.68) suggests the replacement UV-LEDs had the desired effect without the application of
corrections for the HONO conversion efficiency. The slope of ~ 0.91 suggests that the new 385
nm lamp was able to convert the majority of HONO. The discrepancy suggests that ~ 9% of
HONO was converted by the 395 nm lamp. The scatter evident in Fig. 7 at low mixing ratios
may be due to atmospheric dynamic effects resulting in a rapidly changing $NO_2$ background on
timescales faster than the response of the instrument (30 s⁻¹).  A positive 5 pptV positive
intercept indicates a small systematic off-set in the pHONO instrument.





Accuracy and uncertainty in unstable conditions could be improved by measuring at the two different wavelengths concurrently, rather than consecutively. In the same way photolytic $NO_2$ measurement is improved by measuring concurrently with NO, rather than consecutively. This would require three chemiluminescent analysers in parallel, with two photolytic converters. However, in ambient indoor air quality monitoring, where HONO is seen as increasingly important (Gligorovski, 2016), a simple single channel, dual wavelength design might be appropriate and useful.

## 4    Conclusions

An instrument for *in*-situ determination of HONO photolytically has been developed, characterized and deployed in the field as a proof-of-concept. During an atmospheric simulation chamber comparison, the HONO measured corresponded well with FT-IR measurement. During field tests the photolytic HONO instrument agreed reasonably well with the established LOPAP instrument, though the limitations of having a 2-channel sequential measurement were apparent at times; this would be easily overcome in a 3-channel concurrent system. Calibration would gain from a pure HONO source; currently the pHONO calibration requires an independent, direct $NO_2$ measurement and $NO_y$ measurement.

## Acknowledgments

The authors would like to express their gratitude to Dr Marty Buhr or Air Quality Design inc. their support, and Dr Lisa Whalley of Leeds for Spectral Radiometer/calibration equipment. Dr Iusti Bejan, formerly of Leeds, receives thanks for his invaluable guidance in HONO quantification within HIRAC. The financial support of the Engineering and Physical Sciences Research Council (ESPRC) for the studentship of Charlotte Brumby, and the financial support of NCAS, the National Centre for Atmospheric Science, and of NERC, the Natural Environmental Research Council for supporting the studentship of Chris Reed are gratefully acknowledged.

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





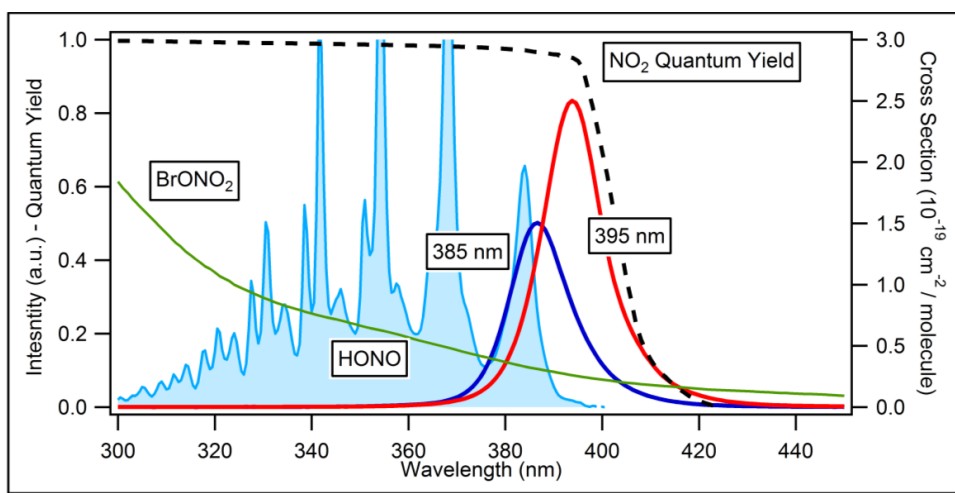

Figure 1. The measured spectral output of two UV-LED elements, nominally 385 nm output in
dark blue, and 395 nm in red. The HONO absorption spectrum is shown in light blue whilst the
NO$_2$ quantum yield is shown in dashed black. The absorption cross section of BrONO$_2$ is shown
in green.





1    Table 1. Showing the distribution of $NO_y$ species NO, $NO_2$, $HNO_3$, and HONO produced from

2    the HONO permeation source.

| # | $NO_y$ ppb Measured | NO ppb Measured | $NO_2$ ppb Measured | $HNO_3$ ppb Calculated | HONO ppb Calculated |
|---|---|---|---|---|---|
| 1 | 20.40 | 3.34 | 2.64 | 0.35 | 14.08 |
| 2 | 19.29 | 2.96 | 2.35 | 0.30 | 13.68 |
| 3 | 16.82 | 2.59 | 2.10 | 0.26 | 11.89 |
| 4 | 14.95 | 2.27 | 1.87 | 0.20 | 10.62 |
| 5 | 13.40 | 2.05 | 1.73 | 0.16 | 9.45 |
| 6 | 12.15 | 1.86 | 1.58 | 0.14 | 8.57 |
| 7 | 11.09 | 1.70 | 1.46 | 0.12 | 7.81 |
| 8 | 10.17 | 1.60 | 1.35 | 0.11 | 7.14 |
| Percent | % | 15.5 | 12.8 | 1.3 | 70.4 |





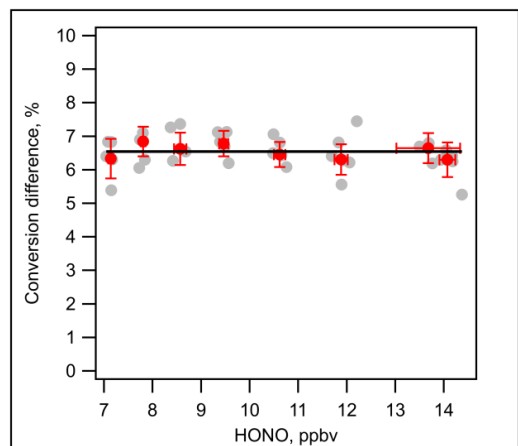

2    Figure 2. Difference in HONO conversion between 385 and 395 nm UV-LEDs over a range of

3    dilutions. Median values are in red, while all data is shown in grey. Linear fit is in black.



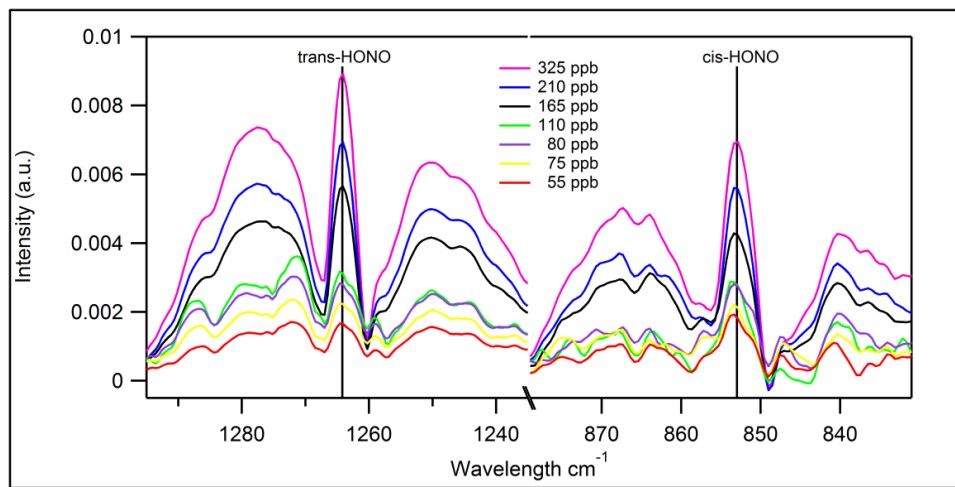

2   Figure 3. FT-IR spectra of dominant HONO absorbance lines at 1264, 853cm$^{-1}$, over a range of

3   concentrations.





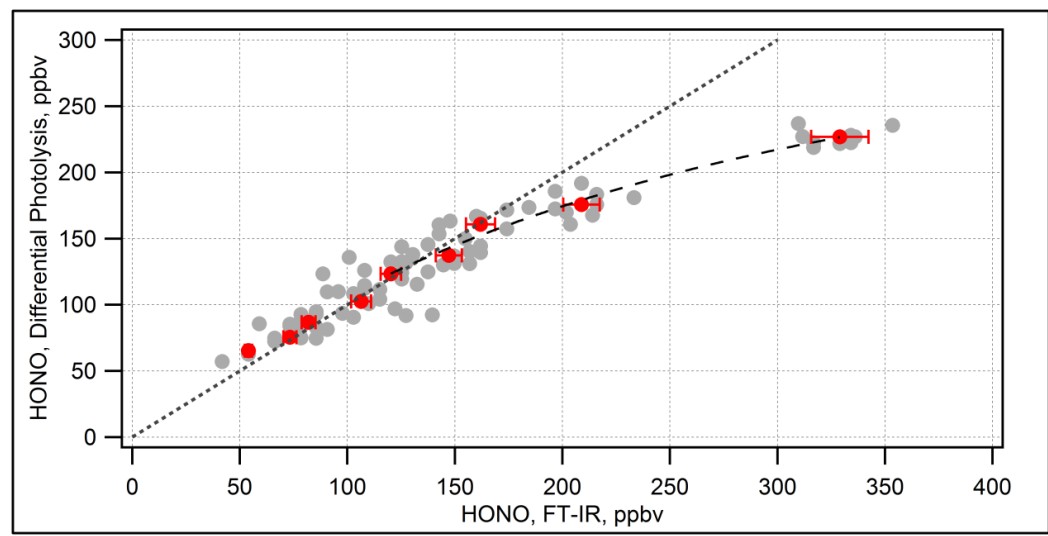

Figure 4. HONO determined by FT-IR (y-axis), versus HONO measured by the
photolytic/chemiluminescence differential photolysis instrument (x-axis). Median values at each
dilution are in red; all values are shown in grey. The 1:1 line is shown for reference as well as an
exponential fit above 150 ppbV HONO.



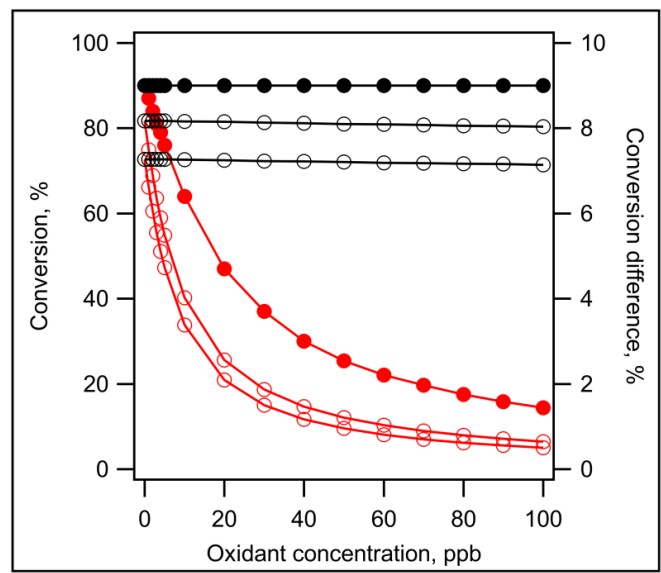

2    Figure 5. Simulated conversion (open circles), and different in conversion (closed circles) for

3    photolytic converters with different j in the presence of OH (red) and $O_3$ (black) oxidants.





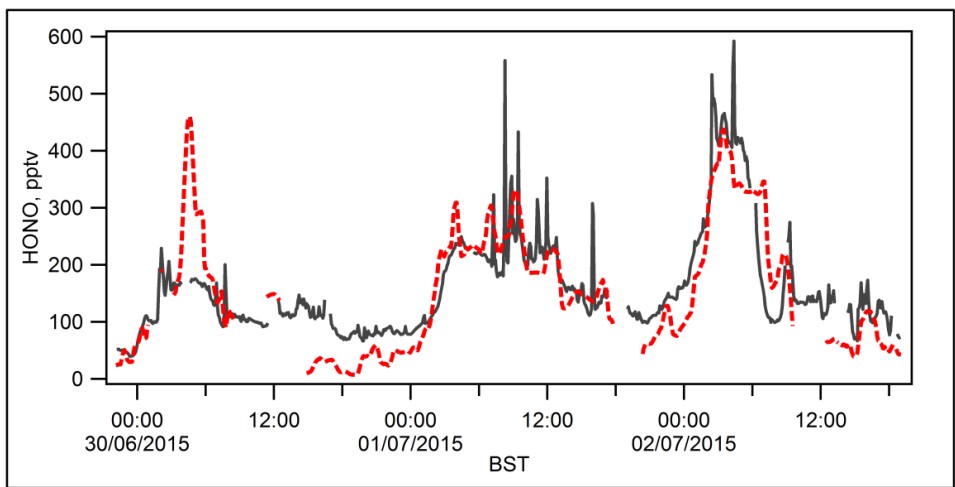

Figure 6. HONO time series during July 2015 at the Weybourne Atmospheric Observatory (WAO) measured by LOPAP (grey) and pHONO (red).



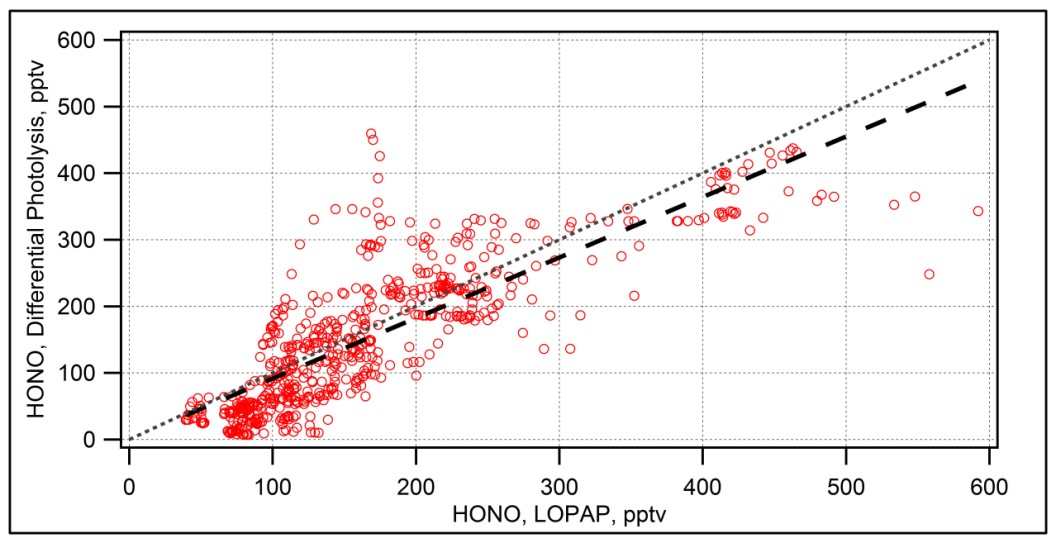

2   Figure 7. Correlation between HONO measured by LOPAP (x-axis) and pHONO (y-axis). The

3   linear correlation is shown in black and the 1:1 line is shown for reference.

