# Peer review of "HONO Measurement by Differential Photolysis"

_Atmospheric Measurement Techniques, 2016_

## Referee Comment (RC1) · Anonymous Referee #1 · 1 Mar 2016

This manuscript describes a clever method of quantifying atmospheric HONO using a chemiluminescence NO analyzer equipped with two photolytic converters with different wavelengths. The material is certainly relevant to the scope of AMT and the methods appear sound and in general well described. I recommend it for publication in AMT after the issues below are addressed.

Major comments: 1. The instrument's HONO measurements are compared to measurements using an FT-IR system at high concentrations. The "absorptivity data" (i.e., IR line strengths or absorption cross sections) were based on an "internal FT-IR cross-section database", as provided by a personal communication. Since these FT-IR data have not been published in the peer-reviewed literature and the methods used to determine the IR line strengths are not described, these nice comparison experiments are just as much a validation of the FT-IR as they are a validation of the present technique.... In other words, the favorable comparison observed is not a \*strong\*

[Figure]

validation of the differential photolysis method. Note that Lee et al. (2012) found large errors (more than a factor of two) in a similar unpublished IR database.

2. The determination of the LOD and precision needs to be more fully described. The text states that the apparent HONO conversion efficiency determines the LOD, and states that the LOD is 40 ppt min-1. As described in equation 1, [HONO] is proportional to the difference between NO2+385 and NO2+395, divided by the difference in HONO conversion efficiencies. The precision is thus determined by the quadrature sum of the two channel's readings. What is the absolute precision (i.e., in ppt NO) of the NOx analyzer's 30 second readings at typical NO + NO2 + HONO concentrations? This would appear to determine the theoretical detection limit. In actual field use, variability of the ambient NO, NO2, and HONO concentrations could limit this precision significantly, as described on pg. 13. What were typical LOD's for the field data? It would be VERY illuminating to include a short time series, at least in the SI, that shows the actual raw NO, NOx+385 and NOx+395 measurements along with the derived HONO concentration – for both the chamber data (calm) and ambient data (occasionally turbulent).

Also, though it is common to state an LOD as xyz "ppt min-1", I recommend more accurately stating it as "xyz ppt with one-minute averaging", since 40 ppt/min does not mean 80 ppt in 2 minutes, etc.

Medium importance comment: The description of how many analyzers are used is confusing. Pg 3 line 9 states that "a dual channel" instrument (singular) is used, but pg. 12 states that "two NO chemiluminescence analyzers operate in parallel with duplicated independent equipment." (plural). Based on this and the rest of 2.1, I initially inferred that there are two dual-channel analyzers, and in each of them NO is continuously measured in one channel and the other channel alternates between "NO + 385 photolysis products" and "NO + 395 photolysis products. Or is there just one dual channel instrument – one channel measures NO and the other alternates between the 385 and 395 nm converters? The answer (the latter) was not apparent until pg. 13 where the field data is described.

Minor comments: Pg 2, line 3, remove "...thought to be...". In addition to the two references provided on vehicular HONO emissions, the authors may wish to include references for more recent HONO emission studies, for example Lee at al 2011 (aircraft and diesel), Rappengluck et al 2013 (on-road vehicles), and Roberts et al 2010 (biomass burning).

Pg 2 lines 17 and 24 – note that QC-TILDAS and the "dual laser – quantum cascade laser" are the same instrument. Probably best to just describe as QC-TILDAS.

Pg 7 line 4: This sentence was confusing: "NO2 was measured directly by CAPS using an EPA certified Teledyne AP T500U, to avoid any HONO interference". It would be good to clarify that CAPS is the technique (from Aerodyne) and that the physical instrument is sold by Teledyne. Otherwise it is confusing to those who are familiar with the CAPS instruments sold directly by Aerodyne. On this note, the authors should actually address potential interference of HONO in the CAPS NO2 measurement since it is based on absorption of light in a bandpass of 440 – 460 nm. Glyoxal is a known interference with the CAPS NO2 measurement. ... What about HONO at the calibration concentrations used?

Pg 8 line 4 – should this be "...apparent differential conversion of 6.54%", instead of "...apparent conversion of 6.54%"?

Figure 6 and 7 and accompanying text: This is an encouraging first set of measurements and comparison for the pHONO instrument, and well described. Any comments on the occasional time periods when the pHONO measures significantly higher than the LOPAP? For example, roughly between 03:00 and 06:00 on 30/6/2015, when pHONO's numbers are 2 to 3x higher?

References Lee, B. H.; et al., "Effective line strengths of trans-nitrous acid near 1275 cm-1and cis-nitrous acid at 1660 cm-1 using cw-QC TILDAS". Journal of Quantitative Spectroscopy & Radiative Transfer 113 (15), 1905-1912 (2012)

Lee, B. H.; et al., "Measurements of Nitrous Acid in Commercial Aircraft Exhaust at the Alternative Aviation Fuel Experiment". Environmental Science & Technology 45 (18), 7648-7654 (2011)

Rappenglück B., et al., (2013) "Radical Precursors and Related Species from Traffic as Observed and Modeled at an Urban Highway Junction", J. Air Waste Manage. Assoc., 63:11, 1270-1286, DOI:10.1080/10962247.2013.822438

Roberts et al., "Measurement of HONO, HNCO, and other inorganic acids by negative-ion proton-transfer chemical-ionization mass spectrometry (NI-PT-CIMS): application to biomass burning emissions", Atmos. Meas. Tech., 3, 981–990, 2010 www.atmos-meas-tech.net/3/981/2010

1. Does the paper address relevant scientific questions within the scope of AMT? YES 2. Does the paper present novel concepts, ideas, tools, or data? YES 3. Are substantial conclusions reached? YES 4. Are the scientific methods and assumptions valid and clearly outli ned? YES with exceptions as noted in my review 5. Are the results sufficient to support the interpretations and conclusions? YES 6. Is the description of experiments and calculations sufficiently complete and precise to allow their reproduction by fellow scientists (traceability of results)? YES 7. Do the authors give proper credit to related work and clearly indicate their own new/original contribution? YES 8. Does the title clearly reflect the contents of the paper? YES 9. Does the abstract provide a concise and complete summary? YES 10. Is the overall presentation well structured and clear? YES 11. Is the language fluent and precise? YES 12. Are mathematical formulae, symbols, abbreviations, and units correctly defined and used? YES 13. Should any parts of the paper (text, formulae, figures, tables) be clarified, reduced, combined, or eliminated? NO 14. Are the number and quality of references appropriate? YES with exception noted in review 15. Is the amount and quality of supplementary material appropriate? NO –exemplary raw data of NO and both NOx* channels should be included

---

## Referee Comment (RC2) · Anonymous Referee #2 · 29 Mar 2016

Review of Reed et al., "HONO measurement by differential photolysis" MS No.: amt-2016-17

General comments.

This is a clearly written paper that describes an interesting and novel approach to measuring atmospheric HONO. This species couples the HOx and NOx cycles and its accurate measurement has been the subject of some controversy.

In principle this is a compelling idea. The paper is clearly written and its logic is easy to follow. The differential photolysis technique described in this paper depends on differencing three numbers (detector response for ambient NO, NO plus some fraction of NO2 photolysed at 385 nm, and NO plus a different fraction of NO2 photolysed at 395 nm plus some fraction of HONO) to accurately derive HONO. Crucially, the chemiluminescence selectivity for NO, the photolytic selectivity for NO2, and the photolytic

selectivity for HONO will determine the atmospheric applicability of this technique. In principle this might be tractable, but in practice this paper does not do a sufficiently thorough job in characterizing the uncertainties in these three measurements. With some additional work this could be suitable for publication, as described below.

This approach may not necessarily require an accurate NO2 measurement in order to measure HONO (although that would greatly simplify the analysis, and seems to be assumed). This approach does require that any interferences in the NO2 measurement at 385 nm are unchanged when HONO is measured at 395 nm. This assumption is implicit, but should be stated explicitly, and defended at some level in the text.

A paper in review at ACPD by these authors shows that undesired thermal decomposition of PAN presents an interference in their NO2 measurement (http://www.atmos-chem-phys-discuss.net/acp-2015-789/). Further, possible interferences from thermal decomposition of methyl pernitrate (CH3O2NO2) (Browne et al., ACP, 11, 4209–4219, doi:10.5194/acp-11-4209-2011, 2011) may also compromise the NO2 measurement in certain regions. Are these an issue for the HONO measurement by difference from NO2 in this approach, and if not, why not?

Further, no discussion is given of the spurious NO2 "artifact" signal that can be generated when illuminating NO2-free air at UV wavelengths (e.g., Kley and McFarland, J. Atmos. Technol., 1980). This NO2 "artifact" signal can be large and variable, and if different at the different wavelengths used to infer HONO in this technique, can represent a major bias in inferred HONO unless accounted for. Are there similar "artifact" signals for HONO when HONO-free air is irradiated at 395 nm? The authors should discuss these issues, quantify them in their instrument, and include them in a comprehensive uncertainty analysis before this paper is acceptable for publication.

Specific comments.

P4, line 7: change to read "...gives a sample residence time of 0.96 seconds assuming plug flow..." How good is the plug flow assumption?

[Figure]

P5, lines 9-10: "It is noted that at both 385 nm and 395 nm there is potential inter-ference from BrONO2 (or in fact any other compounds which photolyse to give NO at either wavelength)..." A point that first comes up here: the potential interferences also include spurious NO chemiluminescence when the photolysis cell is illuminated at either wavelength in the absence of gas-phase nitrogen compounds. This has been termed an "artifact" signal in NO2 photolysis and is thought to arise from undesired photolysis of nitrogen compounds (likely nitric acid or ammonium nitrate, but others are possible) adsorbed on the photolysis cell walls. At times this artifact signal has been substantial (e.g. on the photolytic NO2 measurement flown on the NASA ER-2 aircraft described in Del Negro et al., JGR, 1999) and should not be neglected. I did not find any discussion of the NO2 "artifact", or the possibility of its being different at the different wavelengths used to infer HONO abundance. The "artifact" needs to be quantified fully and its uncertainties discussed as a possible source of error in HONO inferred by this technique.

P8, lines 2-5: "This 2 apparent HONO conversion determines the limit of detection, which is the ability of the analyser to discriminate the difference in signal arising from photolysis at the two different wavelengths from photon counting noise." Because this method is a difference of three numbers (NO, NO2, and HONO) that can vary inde-pendently, the effective limit of detection should also depend on the NO concentration, the NO-to-NO2 ratio, and the NO-to-HONO ratios. For example, when ambient NO is very low (e.g. after dark in remote regions) the 'background' signal (really, ambient NO plus the fraction of NO2 photolysed at 385 nm) from which HONO is determined will be relatively low, and the effective HONO LOD will improve due to better precision in the 'background' signal according to photon counting statistics. Conversely, when am-bient NO is very high (e.g. in urban areas at the surface, close to traffic sources) the 'background' signal can be very high, thus the effective HONO LOD must be degraded. Please include a more rigorous treatment of the HONO limit of detection.

P8, lines 9-11: "The effect of the back reaction of OH + NO, reforming HONO, before

detection of NO, thus reducing the NO signal in the NOx/HONO measurement in the presence of HONO was calculated using a box model..." The ensuing discussion uses a reaction time of 0.11 seconds, which assumes this reaction begins only after the sample exits the photolysis cell. Is it really that simple? There is a similar back-reaction that involves NO + O3P, reforming NO2, which if not accounted for will bias the NO2 measured at 385 nm and thus HONO inferred by this technique. Further, what about HONO and/or NO2 that are photolysed immediately upon entering the photolysis cell? The reaction time to reform either HONO or NO2 can take place in the photolysis cell as well. Both back-reactions, including those occurring in the photolysis cell residence time of about 1 second, should be modeled.

P12, line 23: "...NO offset was taken between these times." What does this mean? Please rephrase for clarity.

P13, lines 1-2: "There is reasonable agreement between the established LOPAP method of HONO measurement 1 and that provided by the pHONO instrument without correction or calibration." This is a value judgement and should be changed to express a quantitive measure of agreement.

P13, lines 3-4: "During 2 the high ozone and high HONO events observed on the 1st and 2nd especially there is very good 3 agreement between the two." Yes, but what about the data observed on June 30 shown in Figure 6? The measurements diverge by a factor ∼2 and lines 21-22 suggest the June 30 data were not plotted in Figure 7. I find the focus on days characterized as "high HONO" to be troubling. The comparison at low values is equally useful and should be included. Please include all the available data in Figures 6 and 7, and explain why a subset was chosen for the linear fit in Figure 7. Is the linear fit single-sided or does it allow for uncertainties in both X and Y values, as it should? Please confirm that a bivariate fit weighted by instrument precisions was used. If at times the two measurements disagree systematically by a factor of two, what does this imply for the accuracy and selectivity of either measurement? It would help to also show the time series of NO and NO2 data in Figure 6, because it is likely the

pHONO measurement will be at its best at relatively low NOx and high HONO values.

P14, lines 11-13: "During 11 field tests the photolytic HONO instrument agreed reasonably well with the established LOPAP 12 instrument,.." This is a value judgement and should be changed to express a quantitive measure of agreement.

Figures.

Fig. 1. For clarity please use colored text to identify the different spectra. The caption describes the colors, but is cut off after line 2 in my PDF version.

Fig. 2. The error bars should show the standard error of the mean – please confirm in the caption.

---

## Author Comment (AC1) · 26 Apr 2016

We would like to thank the reviewer for their comments and suggestions, and for taking the time to review our submission. Please not that supplementary information has been uploaded in addition to this response.

"The instrument's HONO measurements are compared to measurements using an FT-IR system at high concentrations. The "absorptivity data" (i.e., IR line strengths or absorption cross sections) were based on an "internal FT-IR cross-section database", as provided by a personal communication. Since these FT-IR data have not been published in the peer-reviewed literature and the methods used to determine the IR line strengths are not described, these nice comparison experiments are just as much a validation of the FT-IR as they are a validation of the present technique.... In other words, the favorable comparison observed is not a *strong* validation of the differential photolysis method. Note that Lee et al. (2012) found large errors (more than a factor

of two) in a similar unpublished IR database."

Indeed, it is a validation of the Wuppertal cross-section database as much as it is of our instrument. The reviewers reference (Lee et al., 2012) describes different wavenumbers and a different technique (QCL) to that described here (FT-IR).

We have therefore taken the published line strengths found by (Barney et al., (2000) who also found discrepancies between different published values. This has changed our figure only slightly and not affected the point we were showing i.e. that the correlation is linear up to a point, before deviating. The revised FT-IR values using data from (Barney et al., (2000) are $\sim 7\%$ lower, still lie within our measurement uncertainty of $\sim 12\%$ with respect to the 1:1 correlation.

We have also noted that there are several published cross-sections, none of which agree particularly.

"The determination of the LOD and precision needs to be more fully described. The text states that the apparent HONO conversion efficiency determines the LOD, and states that the LOD is 40 ppt min-1. As described in equation 1, [HONO] is proportional to the difference between NO2+385 and NO2+395, divided by the difference in HONO conversion efficiencies. The precision is thus determined by the quadrature sum of the two channel's readings. What is the absolute precision (i.e., in ppt NO) of the NOx analyzer's 30 second readings at typical NO + NO2 + HONO concentrations? This would appear to determine the theoretical detection limit. In actual field use, variability of the ambient NO, NO2, and HONO concentrations could limit this precision significantly, as described on pg. 13. What were typical LOD's for the field data? It would be VERY illuminating to include a short time series, at least in the SI, that shows the actual raw NO, NOx+385 and NOx+395 measurements along with the derived HONO concentration – for both the chamber data (calm) and ambient data (occasionally turbulent)."

The theoretical limit of detection is always determined by the photon counting noise, taken as the 2 sigma standard deviation of the 1hz pre-chamber zero measurement averaged over some time by $1\sqrt{n}$ where n is the number of points which is divided by the sensitivity in counts per second, per ppt, that is in the case of NO, for NO2 and HONO this value must be divided by their respective conversion efficiencies also. The linear response of NO chemiluminescence analysers means that the precision should remain the same at any mixing ratio. In practice the zero noise increases in more polluted (and therefore greater NOx) environments. The LOD, for all intents and purposes, doesn't change for NO, NO2, HONO, however, the uncertainty and precision in HONO is greatly affected by atmospheric variability in NO, NO2, HONO which using a switching channel to determine NO2 and HONO. This proof-of-concept deign demonstrates that all too well, however this is not unique to this instrument, any switched channel instrument i.e. most commercial single channel NOx analyzers suffer this problem. Weybourne proved to be much more turbulent than expected, whilst conditions ultimately led to damage to the instrument.

We excluded data which exceeded 5% variability over 1 minute (Two 30 second cycles). This means that the effective limit of detection ranged from $\sim$ 5ppt to 100 ppt averaged over a minute assuming a differential conversion efficiency of 90%.

We have made available as supplementary information the time series of raw (unprocessed and not interpolated) NO,NO2,HONO measured in the HIRAC chamber over a range of dilutions as a figure, as well as the full time series of ambient data taken at Weybourne as a CSV file.

We have also given more thorough treatment to the determination of LOD and been careful to keep the distinction between LOD, uncertainty, and precision more clear.

"Also, though it is common to state an LOD as xyz "ppt min-1", I recommend more accurately stating it as "xyz ppt with one-minute averaging", since 40 ppt/min does not mean 80 ppt in 2 minutes, etc."

Agreed, we have changed all instances to xyz ppt averaged over 1 minute

"The description of how many analyzers are used is conflfusing. Pg 3 line 9 states that "a dual channel" instrument (singular) is used, but pg. 12 states that "two NO chemiluminescence analyzers operate in parallel with duplicated independent equipment." (plural). Based on this and the rest of 2.1, I initially inferred that there are two dual-channel analyzers, and in each of them NO is continuously measured in one channel and the other channel alternates between "NO + 385 phofltolysis products" and "NO + 395 photolysis products. Or is there just one dual channel instrument – one channel measures NO and the other alternates between the 385 and 395 nm converters? The answer (the latter) was not apparent until pg. 13 where the field data is described."

We have amended the description to make it clear that there is only 1 analyser which has 2 channels (NO and NO2/NOx/HONO). We had described it as "essentially" two analysers operating in parallel, which caused confusion.

"Pg 2, line 3, remove "...thought to be...". In addition to the two references provided on vehicular HONO emissions, the authors may wish to include references for more recent HONO emission studies, for example Lee at al 2011 (aircraft and diesel), Rappengluck et al 2013 (on-road vehicles), and Roberts et al 2010 (biomass burning)."

Agreed, we have added these references at the reviewers suggestion.

"Pg 2 lines 17 and 24 – note that QC-TILDAS and the "dual laser – quantum cascade laser" are the same instrument. Probably best to just describe as QC-TILDAS."

Thank you to the reviewer for pointing out this duplication which has been removed

"Pg 7 line 4: This sentence was confusing: "NO2 was measured directly by CAPS using an EPA certified Teledyne AP T500U, to avoid any HONO interference". It would be good to clarify that CAPS is the technique (from Aerodyne) and that the physical instrument is sold by Teledyne. Otherwise it is confusing to those who are familiar with the CAPS instruments sold directly by Aerodyne. On this note, the authors should

actually address potential interference of HONO in the CAPS NO2 measurement since it is based on absorption of light in a bandpass of 440 – 460 nm. Glyoxal is a known interference with the CAPS NO2 measurement.... What about HONO at the calibration concentrations used?"

The Teledyne instrument is described by the manufacturer as a CAPS NO2 instrument. We were using CAPS as a generic term, but have now made it clear that Aerodyne developed this patented technology. We have added discussion of the possibility of interference in CAPS instruments – from aerosol nitrate (mitigated by a HEPA filter in this instrument). Glyoxal, which absorbs between 400-460 nm, but is absent from the zero air during the calibration of our instrument. HONO absorbs <390nm (Teledyne CAPS operates with a 450nm bandpass) thus HONO is not an interference in this measurement, nor is HNO3, some of which is produced by our HONO source.

"Pg 8 line 4 – should this be "...apparent differential conversion of 6.54%", instead of "...apparent conversion of 6.54%"?"

We have changed added the word "differential" at the reviewers suggestion.

"Figure 6 and 7 and accompanying text: This is an encouraging first set of measurě-ments and comparison for the pHONO instrument, and well described. Any comˇ-ments on the occasional time periods when the pHONO measures significantly higher than the LOPAP? For example, roughly between 03:00 and 06:00 on 30/6/2015, when pHONO's numbers are 2 to 3x higher?"

Indeed the two methods at times disagree greatly, more than the inherent uncertainty of the switching design in turbulent conditions. We believe this is due to a very local strong source, namely the exhaust of the FAGE instrument. The FAGE vents a high flow of percent level NO through what is essentially a vat of sofnofil sorbent, which we believe oxidizes NO to HONO to some extent, so whilst the NO and NO2 is removed, the HONO is not. There were also other possible sources of local HONO e.g. a tractor nearby.

We have added text as to sources of disagreement.

References

Barney, W. S., Wingen, L. M., Lakin, M. J., Brauers, T., Stutz, J. and Finlayson-Pitts, B. J.: Infrared absorption cross-section measurements for nitrous acid (HONO) at room temperature, J. Phys. Chem. A, 104(8), 1692–1699, doi:10.1021/jp9930503, 2000.

Lee, B. H.; et al., "Effective line strengths of trans-nitrous acid near 1275 cm-1and cis-nitrous acid at 1660 cm-1 using cw-QC TILDAS". Journal of Quantitative Spectroscopy & Radiative Transfer 113 (15), 1905-1912 (2012)

Lee, B. H.; et al., "Measurements of Nitrous Acid in Commercial Aircraft Exhaust at the Alternative Aviation Fuel Experiment". Environmental Science & Technology 45 (18), 7648-7654 (2011)

Rappenglück B., et al., (2013) "Radical Precursors and Related Species from Traffic as Observed and Modeled at an Urban Highway Junction", J. Air Waste Manage. Assoc., 63:11, 1270-1286, DOI:10.1080/10962247.2013.822438

Roberts et al., "Measurement of HONO, HNCO, and other inorganic acids by negative-ion proton-transfer chemical-ionization mass spectrometry (NI-PT-CIMS): application to biomass burning emissions", Atmos. Meas. Tech., 3, 981–990, 2010

Please also note the supplement to this comment:
http://www.atmos-meas-tech-discuss.net/amt-2016-17/amt-2016-17-AC1-supplement.zip

---

## Author Comment (AC2) · 26 Apr 2016

We, the authors, would like to thank the reviewer for their comment and time taken in reviewing our submission. Please find our responses, point-by-point below.

"This approach may not necessarily require an accurate NO2 measurement in order to measure HONO (although that would greatly simplify the analysis, and seems to be assumed). This approach does require that any interferences in the NO2 measurement at 385 nm are unchanged when HONO is measured at 395 nm. This assumption is implicit, but should be stated explicitly, and defended at some level in the text."

The technique does require accurately knowing the NO2 conversion efficiency, and thus knowing NO2. The technique does not require knowing NO however at all.

We consider photolytic interference being different at different wavelengths e.g. BrONO2 and calculate the uncertainty to be minimal considering the abundance of

BrONO2. We have expanded the section on measurement artifact, interferences and uncertainties accordingly, being specific about their sources and significance.

"A paper in review at ACPD by these authors shows that undesired thermal decomposition of PAN presents an interference in their NO2 measurement (http://www.atmos-chem-phys-discuss.net/acp-2015-789/). Further, possible interferences from thermal decomposition of methyl pernitrate (CH3O2NO2) (Browne et al., ACP, 11, 4209–4219, doi:10.5194/acp-11-4209-2011, 2011) may also compromise the NO2 measurement in certain regions. Are these an issue for the HONO measurement by difference from NO2 in this approach, and if not, why not?"

Under the reasonable assumption that the sample gas illuminated at 385 and 395nm experiences the same temperature, then thermal interferences only affect the NO2 measurement as at both wavelengths the interference would be equal. Differences in NO2 conversion efficiency introduce some uncertainty in the artifact arising from thermal decomposition of NOy. We would hope that anyone wishing to measure NO2 or HONO using photolytic converters would as a prerequisite eliminated these sources of error as a matter of course.

We have added discussion of the uncertainties arising from artifacts.

"Further, no discussion is given of the spurious NO2 "artifact" signal that can be generated when illuminating NO2-free air at UV wavelengths (e.g., Kley and McFarland, J. Atmos. Technol., 1980). This NO2 "artifact" signal can be large and variable, and if different at the different wavelengths used to infer HONO in this technique, can represent a major bias in inferred HONO unless accounted for. Are there similar "artifact" signals for HONO when HONO-free air is irradiated at 395 nm? The authors should discuss these issues, quantify them in their instrument, and include them in a comprehensive uncertainty analysis before this paper is acceptable for publication."

All data was corrected for NO, NO2 and HONO 'artifact' signals by sampling an overflow of zero air generated from compressed air or BOC BTCA-178 zero air both of

which were subsequently passed through 13x molecular sieve, sofnofil, and activated carbon filters as was oxygen for ozone generation. Artifacts were for NO = 0ppt, NO2=319ppt, HONO=49ppt (319+49), NOy=500ppt. For the CAPS NO2 instrument there is no artifact as zero is referenced to zero air.

In the case of NO when measuring ambient air on campaign the NO artifact was taken as the night time NO value (in answer to another point below).

We have added a discussion of measurement artifacts, their uncertainty and correction for.

"P4, line 7: change to read "...gives a sample residence time of 0.96 seconds assuming plug flow..."How good is the plug flow assumption?"

Gas enters the converter, which is perfectly cylindrical, in an annular ring at one end and exits by annular ring at the other. Response time to 95% of maximum is 1 second or better- indicative of plug flow.

We have added that response to 95% maximum is within 1 second.

"P5, lines 9-10: "It is noted that at both 385 nm and 395 nm there is potential interference from BrONO2 (or in fact any other compounds which photolyse to give NO at either wavelength)..."A point that first comes up here: the potential interferences also include spurious NO chemiluminescence when the photolysis cell is illuminated at either wavelength in the absence of gas-phase nitrogen compounds. This has been termed an "artifact" signal in NO2 photolysis and is thought to arise from undesired photolysis of nitrogen compounds (likely nitric acid or ammonium nitrate, but others are possible) adsorbed on the photolysis cell walls. At times this artifact signal has been substantial (e.g. on the photolytic NO2 measurement flown on the NASA ER-2 aircraft described in Del Negro et al., JGR, 1999) and should not be neglected. I did not find any discussion of the NO2 "artifact", or the possibility of its being different at the different wavelengths used to infer HONO abundance. The "artifact" needs to be

quantified fully and its uncertainties discussed as a possible source of error in HONO inferred by this technique."

We have added a discussion of the artifact correction as noted above in response to a previous comment.

"P8, lines 2-5: "This 2 apparent HONO conversion determines the limit of detection, which is the ability of the analyser to discriminate the difference in signal arising from photolysis at the two different wavelengths from photon counting noise." Because this method is a difference of three numbers (NO, NO2, and HONO) that can vary independently, the effective limit of detection should also depend on the NO concentration, the NO-to-NO2 ratio, and the NO-to-HONO ratios. For example, when ambient NO is very low (e.g. after dark in remote regions) the 'background' signal (really, ambient NO plus the fraction of NO2 photolysed at 385 nm) from which HONO is determined will be relatively low, and the effective HONO LOD will improve due to better precision in the 'background' signal according to photon counting statistics. Conversely, when ambient NO is very high (e.g. in urban areas at the surface, close to traffic sources) the 'background' signal can be very high, thus the effective HONO LOD must be degraded. Please include a more rigorous treatment of the HONO limit of detection."

Agreed, at the request of reviewer #1 also we have given more thorough treatment to the determination of LOD. We have also made better distinction between LOD, uncertainty and precision which was previously unclear.

"P8, lines 9-11: "The effect of the back reaction of OH + NO, reforming HONO, before detection of NO, thus reducing the NO signal in the NOx/HONO measurement in the presence of HONO was calculated using a box model..."The ensuing discussion uses a reaction time of 0.11 seconds, which assumes this reaction begins only after the sample exits the photolysis cell. Is it really that simple? There is a similar back-reaction that involves NO + O3P, reforming NO2, which if not accounted for will bias the NO2 measured at 385 nm and thus HONO inferred by this technique. Further, what about

[Figure]

HONO and/or NO2 that are photolysed immediately upon entering the photolysis cell? The reaction time to reform either HONO or NO2 can take place in the photolysis cell as well. Both back-reactions, including those occurring in the photolysis cell residence time of about 1 second, should be modeled."

The effect of the back reactions within the photolytic converter is what is modelled and shown in figure 5 and its associated discussion. The effect of NO + (O3p) and NO + OH back-reactions are what are calibrated for during NO2 and HONO calibration. The back reaction after exiting the converter (in the absence of UV) was discussed to demonstrate that it does not affect overall sensitivity and is not a source of bias as the [OH] is necessarily different at each wavelength.

We have also made it clear that the input model of the OH back reaction was initiated with the output of a model of the 1s residence time within the photolytic converter while illuminated.

We have clarified the discussion of figure 5 to indicate more precisely what is modeled, whilst also being more specific about what is being calibrated for when we determine the converter efficiency in section 2.4.

"P12, line 23: "...NO offset was taken between these times." What does this mean? Please rephrase for clarity."

That is the NO artifact correction made by assuming it is equivalent to a stable nighttime NO value in remote regions (Lee et al., 2009), away from any source, where NO should be zero in the presence of eg > 15ppb O3.

"P13, lines 1-2: "There is reasonable agreement between the established LOPAP method of HONO measurement 1 and that provided by the pHONO instrument without correction or calibration." This is a value judgement and should be changed to express a quantitive measure of agreement."

We have added the R-squared ($\sim$0.6) value of the correlation between the two measurements.

"P13, lines 3-4: "During 2 the high ozone and high HONO events observed on the 1st and 2nd especially there is very good 3 agreement between the two." Yes, but what about the data observed on June 30 shown in Figure 6? The measurements diverge by a factor ∼2 and lines 21-22 suggest the June 30 data were not plotted in Figure 7. I find the focus on days characterized as "high HONO" to be troubling. The comparison at low values is equally useful and should be included. Please include all the available data in Figures 6 and 7, and explain why a subset was chosen for the linear fit in Figure 7. Is the linear fit single-sided or does it allow for uncertainties in both X and Y values, as it should? Please confirm that a bivariate fit weighted by instrument precisions was used. If at times the two measurements disagree systematically by a factor of two, what does this imply for the accuracy and selectivity of either measurement? It would help to also show the time series of NO and NO2 data in Figure 6, because it is likely the pHONO measurement will be at its best at relatively low NOx and high HONO values."

All data in figure 6 is shown again in figure 7 including the 30th of June. Apologies for the confusion caused by our phrasing which we have now corrected. We were suggesting that there was better agreement on 1st and 2nd , rather than the data only being form the 1st and 2nd. A bivariate fit was used covering all data.

We focused on high HONO days as they fell at the beginning of the campaign when there is certainty that the instrument was operating well. However, as requested by reviewer #1 we will make available the raw unprocessed data, full time series, and NOx data as supplemental information.

High HONO values necessarily occur at times of high NOx so the reviewers' theory that the pHONO measurement would be best at high HONO and low NOx cannot be tested. We have changed figures 6 and 7 to include all the time series of both instruments operating together. which spans 30th June to 6th July. The pHONO instrument operated until 25 July until being damaged by water ingress.

We believe that periods of great disagreement are due to very local contamination – corresponding to periods of high nighttime NO in the case of the 3th June. It is possible that the FAGE exhaust, which is a vat of sofnofil sorbent to scrub the % levels of NO emitted. It is likely that some HONO is evolved by this sorbent and that due to its position both instruments would not necessarily sample the plume.

"P14, lines 11-13: "During 11 field tests the photolytic HONO instrument agreed reaǍň-sonably well with the established LOPAP 12 instrument,.." This is a value judgement and should be changed to express a quantitive measure of agreement."

In line with a previous comment this has been changed to indicate the R-squared value as a measure of agreement.

"Fig. 1. For clarity please use colored text to identify the different spectra. The caption describes the colors, but is cut off after line 2 in my PDF version."

The caption is correct in the online version.

"Fig. 2. The error bars should show the standard error of the mean – please confirm in the caption."

The error bars are the standard deviation of the mean. We have amended the figure caption of this figure (and Fig 4) to clarify this.